# Bifunctional Tm^3+^,Yb^3+^:GdVO_4_@SiO_2_ Core-Shell Nanoparticles in HeLa Cells: Upconversion Luminescence Nanothermometry in the First Biological Window and Biolabelling in the Visible

**DOI:** 10.3390/nano10050993

**Published:** 2020-05-21

**Authors:** Oleksandr Savchuk, Joan Josep Carvajal Marti, Concepción Cascales, Patricia Haro-Gonzalez, Francisco Sanz-Rodríguez, Magdalena Aguilo, Francesc Diaz

**Affiliations:** 1Fisica i Cristalografia de Materials i Nanomaterials (FiCMA-FiCNA)−EMaS, Universitat Rovira I Virgili (URV), Campus Sescelades, Marcelli Domingo 1, E-43007 Tarragona, Spain; oleksandr.savchuk@urv.cat (O.S.); magdalena.aguilo@urv.cat (M.A.); f.diaz@urv.cat (F.D.); 2Instituto de Ciencia de Materiales de Madrid, Calle Sor Juana Ines de la Cruz, Cantoblanco, 28049 Madrid, Spain; ccascales@icmm.csic.es; 3Fluorescence Imaging Group, Departamento de Fisica de Materiales, Facultad de Ciencias, Universidad Autónoma de Madrid, 28049 Madrid, Spain; patricia.haro@uam.es (P.H.-G.); francisco.sanz@uam.es (F.S.-R.); 4Departamento de Biología, Facultad de Ciencias, Campus de Cantoblanco, Universidad Autónoma de Madrid, 28049 Madrid, Spain

**Keywords:** luminescence thermometry, biological windows, biolabeling, core-shell nanoparticles, HeLa cells

## Abstract

The bifunctional possibilities of Tm,Yb:GdVO_4_@SiO_2_ core-shell nanoparticles for temperature sensing by using the near-infrared (NIR)-excited upconversion emissions in the first biological window, and biolabeling through the visible emissions they generate, were investigated. The two emission lines located at 700 and 800 nm, that arise from the thermally coupled ^3^F_2,3_ and ^3^H_4_ energy levels of Tm^3+^, were used to develop a luminescent thermometer, operating through the Fluorescence Intensity Ratio (*FIR*) technique, with a very high thermal relative sensitivity. Moreover, since the inert shell surrounding the luminescent active core allows for dispersal of the nanoparticles in water and biological compatible fluids, we investigated the penetration depth that can be realized in biological tissues with their emissions in the NIR range, achieving a value of 0.8 mm when excited at powers of 50 mW. After their internalization in HeLa cells, a low toxicity was observed and the potentiality for biolabelling in the visible range was demonstrated, which facilitated the identification of the location of the nanoparticles inside the cells, and the temperature determination.

## 1. Introduction

Accurate temperature measurements at the nanoscale are important in many industrial processes, as well as in medicine [1,2]. Sometimes, reaching the object whose temperature has to be measured is difficult or even impossible. In this context, noncontact thermometry can be used. Among the different noncontact thermometry techniques, luminescence thermometry offers high spatial and thermal resolutions [3,4]. By using this technique, it might be possible to detect cancer cells at an early stage of development, just by monitoring the temperature rise in the body due to accelerated metabolic activities in abnormal cells [2]. Moreover, luminescence thermometry has been used to control temperature during photothermal therapy, avoiding over-heating damage [5]. All this has been possible via the development of materials whose luminescent properties strongly depend on small changes of the local temperature, that is, materials with elevated thermal sensitivity.

Lanthanide-doped upconversion nanoparticles, that emit light at shorter wavelengths than the light absorbed, emerged as potential candidates for luminescence thermometry since they possess a large range of emissions, from the ultraviolet (UV) to the near infrared (NIR) [6]. Moreover, they do not show photobleaching effects [7], they can be excited with low-cost NIR laser diodes, they avoid the degradation of the fluorescent material, and they prevent autofluorescence from biological tissues and damages generated by UV radiation, [8].

However, one of the main problems of NIR-excited upconversion in lanthanide-based materials, especially concerning visible emissions, is the low quantum efficiency. This limitation is even more important in nanosized particles due to defects and luminescence quenchers linked to their large surface area. By coating the nanoparticles with a suitable inert shell material, these energy losses can be reduced [9]. The most used approaches consist of coating the luminescent nanoparticles with isostructural crystalline undoped materials, and amorphous SiO_2_. An additional advantage of using this last approach is that the SiO_2_ shell increases the hydrophilicity of the nanoparticles. This makes their dispersion in biological compatible fluids easier, and protects them from the luminescence quenching caused by differences in pH and calcium levels in biological tissues and living cells [10]. On the other hand, we recently demonstrated the protection from over-heating during the excitation process offered by the silica shell, as well as the enhancement of the thermal sensitivity of the luminescent thermometers [11].

The visible upconversion emissions used for ratiometric luminescence thermometry have a limited penetration depth in biological tissues because of light scattering. Thus, if lanthanide-based nanoparticles are to be used for deep subcutaneous bioimaging, they should emit in the three transparency windows of biological tissues, the so-called biological windows (BW) [12]. The first BW (I-BW) lays in the range 650–950 nm, where the optical absorption of biological tissues, and especially the optical absorption of water, is minimized [13]. The second BW (II-BW) lays in the range 1000–1350 nm, where the Rayleigh and Mie scatterings are reduced [14]. Finally, the third BW (III-BW), that has been explored more recently, lays in the range 1500–1800 nm [15], or the more extended 1350–2300 nm one [16,17], depending on the sources. This last biological window is characterized by reduced tissue absorbance and scattering. This allows for the better transmittance of light through oxygenated blood and melanin-containing tissues [16].

The potentiality of using NIR-to-NIR upconversion emissions of Tm^3+^-doped fluoride nanophosphors has been reported, which enabled high cellular contrast and tissue imaging in the I-BW [18]. Additionally, the NIR-excited ^3^F_2,3_→^3^H_6_ and ^3^H_4_→^3^H_6_ upconversion emissions of Tm^3+^, lying in the I-BW and arising from thermally coupled energy levels, seem to be a good choice to develop highly sensitive ratiometric luminescent thermometers [19].

Cell thermometry using luminescent materials has been almost exclusively demonstrated using visible light [20]. This is due to the fact that the penetration depth that can be achieved with visible light in in vitro cell culture is enough to analyze the whole thickness of the sample. Furthermore, another advantage of using this visible light is that it can be easily visualized using conventional optical microscopes, thus it is easy to identify where these nanoparticles are located, record their spectra, and from them, determine the temperature inside of the cell. However, by shifting the emissions to the I-BW, higher thermal sensitivities can be achieved [17], and thermal maps with a clearer spatial resolution can be recorded, due to the lower light scattering in this spectral region [16]. Luminescence thermometry in the I-BW, to determine the temperature inside of living cells, has been reported only once. Lipoic acid-protected Au nanoclusters luminescent thermometers, 1.6 nm in diameter, have been internalized in HeLa cells, and their internal temperature has been determined by fluorescence lifetime thermometry through a time-correlated, single-photon counting technique [21]. Through this approach, the authors determined that the lifetime of the Au nanoclusters was longer inside the cells than in the buffer solution, due to the formation of biomolecular corona surrounding the internalized nanoclusters that modify their physical properties. Thus, after all, it was not clear if the temperature measured inside the cells was the correct one, and the thermal resolution that could be achieved was not higher than the one obtained using luminescent nanothermometers operating in the visible range.

As pointed out above, fluorescent labelling is a widely used and indispensable tool in biology. In this context, lanthanide-doped up-converting nanoparticles, with emissions in the visible range, allow us to overcome some of the disadvantages faced by conventional bio-labelers excited in the UV or in the deep blue region of the electromagnetic spectrum. Among the advantages introduced by lanthanide-doped up-converting nanoparticles, it is important to note here very low autofluorescence, absence of photo damage to living organisms, high detection sensitivity and high depth of light penetration. All these properties make them ideal fluorescent labels for bioimaging [22,23,24,25].

Among the different host matrices for Ln^3+^ ions, GdVO_4_ is characterized by high optical absorption and emission cross-section values for the Ln^3+^-doping ions, high thermal stability, and moderate cut-off phonon energy (700 cm^−1^) [26].

Here we present a study of the temperature-dependent, NIR-excited upconversion emissions lying in the I-BW, grounding the development of a luminescent thermometer operating in this spectral region, consisting of Tm,Yb:GdVO_4_@SiO_2_ core-shell nanoparticles prepared by a soft hydrothermal route, that could also be used as up-conversion biolabels operating in the visible range. Different concentrations of Tm^3+^ as the active ion were studied in order to optimize thermal sensing. Internalization in HeLa cells, which is facilitated by the inert silica shell, was analyzed, demonstrating a low cytotoxicity. The characteristic Tm^3+^ blue emission of these nanoparticles was visualized through photoluminescence microscopy. This emission can be used for biolabeling purposes since it allows easy identification of the location of the particles inside HeLa cells. This facilitated the identification of the region in which the temperature inside the cells could be measured using the luminescent thermometer, developed with these core-shell nanoparticles, operating in the I-BW. The results presented prove the potentiality of Tm,Yb:GdVO_4_@SiO_2_ core-shell nanoparticles as multifunctional platforms, with applications in NIR-to-NIR upconversion temperature-sensing in the I-BW with high thermal sensitivity, and NIR-to-visible upconversion biolabeling.

## 2. Materials and Methods

### 2.1. Synthesis of Tm,Yb:GdVO_4_@SiO_2_ Core-Shell Nanoparticles

Gd_0.85−*x*_Yb_0.15_Tm*_x_*VO_4_ (*x* = 0.5–3 mol%) nanoparticles were prepared following a soft hydrothermal synthesis method. Starting reagents consisted of Gd_2_O_3_, Yb_2_O_3_, Tm_2_O_3_ (Strem Chemicals, Newburyport, MA, USA, 99.9%) and NH_4_VO_3_ (Sigma Aldrich, St. Louis, MO, USA, 99%). We prepared Ln^3+^-nitrates by dissolving the required amounts of the corresponding oxides in a nitric acid solution (50% distilled H_2_O:50% HNO_3_ at 69%), and heating them until complete dryness. Then, we dissolved these Ln^3+^-nitrates in 10 mL of distilled H_2_O and added them to a solution previously prepared of NH_4_VO_3_ in 20 mL of distilled water. We then adjusted the pH of the resulting solution to 7 with diluted NH_4_OH. After 15 min of magnetic stirring, the dispersion formed was heated in a Teflon-lined autoclave at 458 K for 24 h. The resulting product was collected by centrifugation, washed with distilled water several times, and dried overnight at 393 K. Finally, the samples were annealed at 873 K for 5 h to remove surface defects formed during the low-temperature synthesis process and to promote a higher degree of crystallization [27]. With the aim of achieving a further improvement of the optical emission efficiency of these nanoparticles, they were coated with a SiO_2_ layer with a thickness of 5–7 nm. The coating process was performed by dispersing the Gd_0.85−*x*_Yb_0.15_Tm_*x*_VO_4_ nanoparticles (1.5 mmol) in a solution formed by 40 mL of ethanol (99.5% Emplura Merck, Darmstad, Germany) and 4 mL of distilled H_2_O, to which we added 0.75 mL of NH_4_OH to adjust the pH to 9, and 2 mL of tetraethoxysilane [Si(OC_2_H_5_)_4_, TEOS, Alfa Aesar, Ward Hill, MA, USA, 99%] slowly added under stirring. The reaction was continued for 4 h, and the final product was washed with ethanol, centrifuged and dried at 423 K.

### 2.2. Structural and Morphological Characterization

Powder X-ray diffraction (XRD) was performed on Tm,Yb:GdVO_4_ and Tm,Yb:GdVO_4_@SiO_2_ core-shell nanoparticles. For that a Bruker (Billerica, MA, USA) AXS D-8 Advance diffractometer, using Cu Kα radiation and operating at room temperature, was used.

The morphology of the nanoparticles was characterized by conventional and high-resolution transmission electron microscopy (TEM and HRTEM, respectively) with JEOL (Tokyo, Japan) JEM2100 and JEOL JEM3000F microscopes, operating at 200 kV and 300 kV accelerating voltages, respectively.

Hydrodynamic particle size distribution was measured by dynamic light scattering (DLS, Vasco 2, Cordouan Technologies, Pessac, France), and Fourier transform infrared absorption (FT-IR, Nicolet 20SXC, Nicolet Instruments Corp., Madison, WI, USA) spectra was used to investigate the formation of the Tm,Yb:GdVO_4_@SiO_2_ structure and the presence of adsorbed species on the surface of the nanoparticles. These results can be consulted elsewhere [27].

### 2.3. Temperature-Dependent Photoluminescence Measurements

Tm,Yb:GdVO_4_@SiO_2_ core-shell nanoparticles were placed in a Linkam (Tadworth, UK) THMS 600 heating stage for temperature dependent photoluminescence measurements. They were excited by a fiber-coupled laser diode with emission at 980 nm and a power of 50 mW. The laser beam was focused on the sample by a 40× microscope objective with N.A. = 0.6 providing a spot size of around 10 µm, limiting the excitation density to a value for which we did not observe local thermal effects caused by the laser, and which did not affect the temperature reading measurements by introducing artifacts. The emission signal was collected by the same microscope objective, and after passing a dichroic filter to eliminate the excitation wavelength, was sent to an AVANTES (Apeldoorn, The Netherlands) AVS-USB2000 fiberoptic spectrometer to record it.

### 2.4. Subtissue Spectroscopic Measurements

The subtissue penetration depth that could be achieved by using the Tm,Yb:GdVO_4_@SiO_2_ core-shell nanoparticles was investigated by using a double beam fluorescence microscope. This experimental set-up allowed us to place a phantom tissue of variable thickness between the Tm,Yb:GdVO_4_@SiO_2_ core-shell nanoparticles and the detection optics. The penetration depth was obtained by monitoring the infrared luminescence generated at different tissue thicknesses in Intralipid as phantom tissue. Intralipid is an absorbing and scattering medium that has been extensively used in the past to mimic the optical properties of human skin in the I- and II-BWs. In this work we used Intralipid 10%, diluted to a concentration of 2%. At this concentration, the wavelength dependence is similar to that previously reported for several human tissues [28].

### 2.5. In Vitro Experiments

HeLa (Human cervical cancer cell line) cells were grown as a monolayer, employing Dulbecco’s Modified Eagle Medium (DMEM) (Sigma Aldrich, St. Louis, MO, USA) containing 10% fetal calf serum (FCS), 50 units mL^−1^ penicillin and 50 μg mL^−1^ streptomycin (Sigma Aldrich, St. Louis, MO, USA). The cells were incubated at 310 K in a humidified 5% CO_2_ atmosphere. The medium was changed daily. For the fluorescence observation, cells were placed onto round coverslips placed into 24 wells plates.

For evaluation of Tm,Yb:GdVO_4_@SiO_2_ core-shell nanoparticles on an in vitro thermal scanning experiment, the cells were seeded in 24 well plates containing sterile round coverslips and were kept in a sterile environment in the incubator for 24 h until the experiment started. After 2 h of incubation with Tm,Yb:GdVO_4_@SiO_2_ nanoparticles, the cells were washed with phosphate-buffered saline (PBS) solution, and used in luminescence experiments.

### 2.6. In Vitro Cell Cytotoxicity/Viability Studies

To determine cell cytotoxicity/viability, the cells were placed in a 24 well plate at 310 K in 5% CO_2_ atmosphere. After 48 h of culture, the medium in the well was replaced with a fresh medium containing the Tm,Yb:GdVO_4_@SiO_2_ nanoparticles in a volume ratio ranging from 1:50 to 1:500, and cells were incubated for 2 h. After incubation, the medium was removed and a new complete medium without nanoparticles was added. After 24 h, 0.5 mL of 3-[4,5-dimethylthiazol-2-yl]-2,5-diphenyltetrazolium-bromide (MTT) dye solution (0.05 mg mL^−1^ of MTT, Sigma Aldrich, St. Louis, MO, USA) was added to each well. After 2–3 h of incubation at 310 K and 5% CO_2_, the medium was removed, and formazan crystals were solubilized with 0.5 mL of dimethylsulphoxide (DMSO). The solution was vigorously mixed to dissolve the reacted dye. The absorbance of each well was read at 540 nm on a Spectra Fluor4 (Tecan Group Ltd., Männedorf, Switzerland) microplate reader. The spectrophotometer was calibrated to zero absorbance, using culture medium without cells. The relative cell viability (%) related to control wells containing cell culture medium without nanoparticles was calculated by [A]_test_/[A]_control_ × 100, where [A]_test_ is the absorbance of the test sample and [A]_control_ is the absorbance of the control sample. The MTT assay is a simple non-radioactive colorimetric assay to measure cell cytotoxicity, proliferation or viability. MTT is a yellow, water soluble, tetrazolium salt. Metabolically active cells are able to convert this dye into a water-insoluble, dark blue formazan, by reductive cleavage of the tetrazolium ring [29]. Formazan crystals can then be dissolved in an organic solvent such as DMSO, and quantified by measuring the absorbance of the solution at 540 nm, and the resultant value is related to the number of living cells.

## 3. Results

All pale-yellow products obtained after the hydrothermal reaction were shown to be isostructural to the tetragonal *I*4_1_/*amd* zircon-type phase of GdVO_4_ (JCPDS File 86-0996), indicating a 100% synthesis yield. The crystal phase was maintained after 5 h annealing at 873 K, although with narrower Bragg peaks. This indicates an increase of the average size of individual crystalline domains that constitute vanadate nanoparticles. No additional reflections were detected for SiO_2_-coated samples, apart from a rougher background reflecting the presence of amorphous SiO_2_, as compared to the XRD of the bare Tm,Yb:GdVO_4_ annealed nanoparticles (NPs), included in Figure 1a for comparison. The synthesis was repeated several times, resulting always in the same product with reproducible properties.

Figure 1b shows a characteristic TEM image of an annealed sample of Tm,Yb:GdVO_4_ NPs, which presents polygonal forms, mainly of square or rectangular sections, and lengths of 25–40 nm. Figure 1c,d display HRTEM images of Tm,Yb:GdVO_4_@SiO_2_ NPs, with darker and lighter parts being the Tm,Yb:GdVO_4_ core and the amorphous silica coating, respectively. We determined that the coating layer had an average thickness of ~7 nm around the NPs. HRTEM images of discrete nanoparticles reveal a well-defined crystalline structure in the core, with a lattice fringe distance of 0.267 nm, as can be seen in Figure 1d, matching the (112) interplanar spacing of GdVO_4_, according to the JCPDS File 86-0996.

Figure 2a shows a schematic representation of the Tm,Yb:GdVO_4_@SiO_2_ core-shell nanoparticles to illustrate their potential use as biolabels (operating in the visible range) and luminescence thermometers (operating in the I-BW). Figure 2b shows the upconversion emission spectra of 1% Tm^3+^, 15% Yb^3+^:GdVO_4_@SiO_2_ core-shell nanoparticles, under 980 nm excitation at room temperature and 333 K. In the figure, we indicate the spectral range corresponding to the I-BW in yellow. As can be seen, when the temperature increased, all emission bands drop in intensity, with the exception of the band at 700 nm that slightly increases its intensity. In order to explain why this is happening, first it is necessary to identify the energy levels assigned to each of the radiative transitions. Figure 2c shows the energy level diagram of the Tm^3+^ and Yb^3+^ ions, indicating the pathways for the 980 nm excited upconversion process, and the transitions involved in the generation of such spectra. In a first step, the excitation at 980 nm is absorbed by Yb^3+^ promoting its electrons from the ^2^F_7/2_ fundamental state to the ^2^F_5/2_ excited state. Then, part of this energy is transferred to the ^3^H_5_ energy level of Tm^3+^, from which electrons relax very fast to the ^3^F_4_ energy level. Then, a second energy transfer from Yb^3+^ promotes Tm^3+^ electrons to the ^3^F_2_ energy level, which relaxes, populating the ^3^F_3_ and ^3^H_4_ energy levels. Finally, a third energy transfer process promotes Tm^3+^ electrons in the ^3^H_4_ level to the ^1^G_4_ energy level, from which the blue and red emissions, centered at 475 and 650 nm, arise through radiative ^1^G_4_ → ^3^H_6_ and ^1^G_4_ → ^3^F_4_ transitions, respectively. The emissions at 700 nm and 800 nm arise from the ^3^F_3_ → ^3^H_6_ and ^3^H_4_ → ^3^H_6_ radiative transitions. Along with the non-radiative relaxation from the ^3^F_2_ level, Tm–Tm cross-relaxation processes have been previously reported which involve depopulation of the ^1^G_4_ level and population of the ^3^F_3_ and ^3^H_4_ levels. These mechanisms are favored as Tm^3+^ concentration and temperature increase [30]. Furthermore, a non-radiative decay after the second energy transfer from the ^2^F_5/2_ level of Yb^3+^ allows for populating the ^3^H_4_ level of Tm^3+^ very efficiently, since its energy gap (~3400 cm^−1^) matches the phonon energy of OH^-^ groups adsorbed on the silica surface [27]. This privileged population of the ^3^H_4_ level explains the strong NIR emission intensity observed at 800 nm.

The energy gap between the ^3^F_3_ and ^3^H_4_ energy levels is 2046 cm^−1^ [31]. The relative low energy difference between these two electronic levels allows the existence of a thermal equilibrium between their electronic populations governed by the Boltzmann law. Thus, we can consider that the ^3^F_3_ and ^3^H_4_ energy levels are thermally coupled, and can be used for temperature determination using the fluorescence intensity ratio (*FIR*) technique [32]. Figure 2b shows how the intensity of the peak lying at 700 nm, and arising from the ^3^F_3_ → ^3^H_6_ transition, increases as the temperature increases, while the intensity of the peak lying at 800 nm, and arising from the ^3^H_4_ → ^3^H_6_ transition, decreases as the temperature increases. This observation would prove the thermal coupling between the ^3^F_3_ and ^3^H_4_ energy levels.

Since upconversion luminescence is a non-linear process, the intensity of the emission bands depends on the dopant concentrations, as well as on temperature. To optimize the intensity ratio between the emission lines at 700 and 800 nm, and to obtain the maximum thermal sensitivity, we analyzed the evolution of the *FIR* with temperature for the Tm,Yb:GdVO_4_@SiO_2_ core-shell nanoparticles doped with different concentrations of Tm^3+^. In a previous study we optimized the concentration of Yb^3+^ (~15 mol%) to maximize the intensity of the Tm^3+^ emissions in these core-shell nanoparticles [27], thus in the present work the concentration of Yb^3+^ was kept constant at 15 mol%. The normalized *FIR* between the emission lines at 700 and 800 nm of the *x* mol% Tm, 15 mol% Yb:GdVO_4_@SiO_2_ core-shell nanoparticles, with *x* = 0.5–3 mol%, is shown in Figure 3a. The experimental points were fitted to a Bolzmann distribution equation with an additional offset that takes into account the partial overlapping of the emission lines as stated in Equation (1) [32]:(1)$$FIR=Aexp\left(-\frac{\Delta E}{kT}\right)+C$$
where Δ*E* is the energy gap between the two thermally coupled levels, *k* is the Bolzmann constant, *T* is the absolute temperature, and *A* and *C* are fitting constants. As can be seen in Figure 3a, the core-shell nanoparticles with a 1 mol% Tm^3+^ showed the highest slope.

The absolute thermal sensitivity can be calculated from the first derivative of the *FIR* fittings with respect to the temperature, and might be used for comparison of the same material with different concentrations of dopants [32], as expressed in Equation (2):(2)$$S_{abs}=\frac{\partial FIR}{\partial T}=Aexp\left(\frac{-\Delta E}{kT}\right)\left(\frac{\Delta E}{kT^{2}}\right)$$

The thermal sensitivities calculated for the Tm,Yb:GdVO_4_@SiO_2_ core-shell nanoparticles are presented in Figure 3b. As can be seen, and as expected from the slopes of *FIR* obtained in Figure 3a, the nanoparticles with 1 mol% Tm^3+^ exhibit the highest absolute thermal sensitivity, with a maximum of 0.037 K^−1^ at 333 K.

Table 1 shows the comparison of the results we obtained with those reported previously for other Tm^3+^-doped systems using the same intensity ratio. As can be seen, the absolute thermal sensitivity of Tm,Yb:GdVO_4_@SiO_2_ core-shell nanoparticles is the highest among Tm^3+^, Yb^3+^ co-doped systems, being one or two orders of magnitude higher than the *S_abs_* reported for most of the other luminescent thermometers. The only exception is Tm,Yb:YAG fiber optics [33], with a *S_abs_* of the same order of magnitude, but still 76% smaller than the one obtained in this work, being the highest ever reported for Tm^3+^, Yb^3+^ co-doped systems operating in the first biological window.

The better performance of our core-shell nanoparticles in terms of thermal sensitivity might be related to the SiO_2_ inert shell layer, as we showed previously in Er,Yb:GdVO_4_@SiO_2_ nanoparticles [11]. The inert shell structure preserves the luminescence generated by the active core, and prevents interactions with solvent molecules that might quench the emissions of Tm^3+^. It also avoids the progressive heating of the nanoparticles when exposed to extended excitation periods with the laser emitting at 980 nm, and preserves their detrimental effects for thermometric applications [11].

Although *S_abs_* is useful for comparing luminescent thermometers operating under the same conditions, the relative thermal sensitivity (*S_rel_*) allows for the comparison with other systems, operating in different spectral ranges and under different conditions. The relative thermal sensitivity is defined as [1] stated in Equation (3):(3)$$S_{rel}=\frac{1}{FIR}\left|\frac{\partial FIR}{\partial T}\right|=\frac{FIR-C}{FIR}\frac{\Delta E}{kT^{2}}$$

Figure 3c shows *S_rel_* calculated for our particles, with a value of 3.3% K^−1^ at 298 K. Since it depends on the *ΔE* value, that does not change substantially when changing the doping concentration, only a single value can be given for all the nanoparticles analyzed. The *S_rel_* values reported in the literature are between 1.0 and 28% K^−1^, with most of the values being in the range 2.0–3.3% K^−1^ [34,35,36,37,38,39,40]. The value we report here for Tm,Yb:GdVO_4_@SiO_2_ nanoparticles is the highest reported up to now in the literature for Tm^3+^-doped systems, only surpassed by that of Tm:NaYbF_4_@SiO_2_ nanoparticles, although in that case, the maximum thermal relative sensitivity has been found at 100 K, far away from the biological range of temperatures. In fact, *S_rel_*, when determined at RT, takes a similar value to those reported for the rest of materials listed in Table 1. A higher value was reported for Tm,Yb:Y_2_O_3_ sub-micron size particles, although in this last case, the electronic levels from which the emission bands to be used as luminescent thermometers arise are not thermally coupled, and thus a modification of the *FIR* model has to be used [41]. This does not allow for a direct comparison with the rest of the materials included in the table, since they do not operate purely in the I-BW.

The temperature resolution can be calculated from *S_rel_* as stated in Equation (4) [20]:(4)$$\delta T=\frac{1}{S_{rel}}\frac{\delta FIR}{FIR}$$
where *δFIR/FIR* is the relative uncertainty in the determination of the thermometric parameter, *FIR* in our case, and depends on the acquisition setup. Here, we use the typical value of 0.5% [13].

Figure 3c also shows *δT* for our particles. Table 1 also lists the *δT* values reported in the literature or calculated by us from the data reported in previous publications. As can be seen, the value of the temperature resolution we report for Tm,Yb:GdVO_4_@SiO_2_ nanoparticles is among the smallest reported for Tm^3+^-doped systems, although we found several materials exhibiting a similar temperature resolution value, from 0.16 to 0.18 K, including Tm,Yb:LiNbO_3_ single crystals, and Tm,Yb:YF_3_ microcrystals. However, if we take into account the size of the material, the *δT* value we report is the smallest for Tm^3+^-doped nanoparticles operating in the I-BW.

Finally, to assess the performance of our luminescent nanothermometers, we determined their reproducibility, i.e., the change of the same measurement carried out under modified circumstances. For that, we recorded the spectra and determined *FIR* in 10 heating–cooling cycles between 298 and 310 K. This reproducibility is higher than 99%, as can be seen in Figure 3d, computed as stated in Equation (5) [20]:(5)$$R=1-\frac{max\left[FIR_{c}-FIR_{i}\right]}{FIR_{c}}$$
where *FIR_c_* is the *FIR* mean value, and *FIR_i_* is the value of *FIR* measured at each temperature.

Motivated by the good results of the thermal sensitivity and temperature resolution obtained, we analyzed the potentiality of using Tm,Yb:GdVO_4_@SiO_2_ core-shell nanoparticles in biological systems. To analyze how the fluorescence intensities of the Tm^3+^ luminescence bands were attenuated by the biological tissue, we used a phantom tissue of variable thickness. A drop of a dispersion of the core-shell nanoparticles in PBS was placed in a microscope slide, and was optically excited by a continuous fiber-coupled diode laser at 980 nm. The laser beam was focused on the core-shell nanoparticles using a 20× microscope objective (N.A. = 0.4). The same objective was used to collect the Tm^3+^ emission. After passing a dichroic mirror to eliminate the excitation radiation, the emission at 800 nm was analyzed using a highly sensitive Si CCD camera (Synapse, Horiba, Kyoto, Japan) attached to a high-resolution monochromator (iHR320, Horiba, Kyoto, Japan). Intralipid 10%, diluted in a concentration of 2% in water, was used as phantom tissue.

Figure 4 shows the collected emission intensity for the Tm^3+^ emission band as a function of the tissue thickness. To determine the penetration depth, phantom tissue slices of different thicknesses were placed between the focusing/collecting microscope objective and the Tm,Yb:GdVO_4_@SiO_2_ core-shell nanoparticles dispersion. As can be seen, a monotonous decrease of the collected signal with the tissue thickness is observed, indicating that the emission band of Tm^3+^ is attenuated within the phantom tissue either by absorption or scattering processes (or by a combination of them) [42]. The results show that a significant fluorescence signal could still be obtained up to a tissue thickness of 0.8 mm, defining this as the maximum penetration depth that can be achieved in this case. The results are smaller to those obtained in Tm,Yb:CaF_2_ nanoparticles (maximum penetration depth of 2 mm) using emission lines in the same spectral region [18], probably because fluorides are more efficient emitters when upconversion processes are considered, due to their lower phonon energies [43,44]. However, we believe that this penetration depth might be increased by using a detection set-up with a higher sensitivity, and/or by increasing the power of the excitation radiation, which was limited to 50 mW in these experiments. The excitation power value used is still three orders of magnitude smaller than the one that would cause laser-induced local heating in water solutions [45]. Thus, this leaves room to achieve deeper penetration depths by increasing the excitation power, without causing damage to living biological tissues. However, we limited the power of the excitation radiation, in this case to 50 mW, since this is the power with which incubated HeLa cells were illuminated in the following experiments. Thus, these results demonstrate that the penetration depth achieved with this power is enough for the temperature measurements inside the cells.

For in vitro experiments, we incubated HeLa cancer cells with Tm,Yb:GdVO_4_@SiO_2_ core-shell nanoparticles, as described in the Experimental Section. After incubation, the HeLa cells were excited in a confocal microscope illuminated with a continuous fiber-coupled diode laser at 980 nm. The excitation power was limited at 50 mW, and the spot size on the sample was increased to 3 mm to avoid damages to the living cells. The fluorescence was detected by the same system described before. Figure 5a shows an optical transmission microscope image of the HeLa cells after incubation. The arrow indicates the position of the nanoparticles when the excitation laser is off. Figure 5b shows an optical transmission microscope image of the same region when the excitation laser is on. The blue emission arising from the Tm,Yb:GdVO_4_@SiO_2_ core-shell nanoparticles can be seen. This suggests that the nanoparticles are incorporated into vesicles. This was verified by the intensity recorded for different nanoparticles, getting a higher and clearer image for those located outside of the cells, as can be seen in Appendix A. An enlarged image of the area in which the nanoparticle is located is shown also in Figure 5c. Here, some tiny signs of autofluorescence arising from the cell seem to be present, however, they do not prevent the observation of the emission arising from the luminescent nanoparticle. We performed an intensity scan of the blue emission along the line indicated in Figure 5c to corroborate that it was only arising from the luminescent nanoparticles. The results show an increase of the intensity in the area where the nanoparticles are located, but not in the rest of the cell. The red dashed line in the graph indicates the level of noise. These results demonstrate that the Tm,Yb:GdVO_4_@SiO_2_ core-shell nanoparticles can be used as effective biolabels for HeLa cells in bioimaging experiments through an upconversion mechanism.

The visualization of the location of the luminescent nanoparticles through the emission of visible light allowed us to simplify the task of measuring the temperature inside the cells through the NIR emissions generated by the same core-shell nanoparticles. A characteristic spectrum in this region is shown in Appendix A. The temperature that could be extracted from these spectra, by taking into account Equation (1) and the fitting function result in Figure 3a, was 308 ± 2 K after 5 measurements, coinciding with the incubation temperature of the HeLa cells. The spot of 3 mm used under the excitation conditions to avoid photo damages in the living cells did not allow for higher resolution temperature measurements.

At this point, we should note that we performed a number of different toxicity assays. We found that incubation of HeLa cells with medium solutions containing Tm,Yb:GdVO_4_@SiO_2_ core-shell nanoparticles caused a reduced toxicity, as is shown in Figure 5d, for the different concentrations analyzed.

These results prove that Tm,Yb: GdVO_4_@SiO_2_ core-shell nanoparticles can be used as thermal probes operating in the I-BW, with very good thermal sensitivity and temperature resolution, and with possible use in determining the temperature inside living HeLa cells.

Table 2 shows the lanthanide-doped materials that have been used for cell thermometry up to now, to the best of our knowledge. As can be seen, a very reduced number of articles have been published with this aim. Most of them are based on metallic complexes with organic molecules [46,47,48,49], pumped in the UV range, with the subsequent autofluorescence generation in living cells. Only a couple of them used luminescent nanoparticles of NaYF_4_ doped with Er^3+^ and Yb^3+^ ions, that can be pumped in the NIR at 980 nm, while still emitting in the visible region, especially in the green and red regions [50,51]. However, the thermal resolution that can be achieved with these materials in living cells has not been reported.

Thus, there is still room to develop new luminescent materials that can be used as luminescent nanothermometers to determine temperature inside living cells, especially when exploring new spectral areas like the BWs, which are almost unexplored within this context. Thus, the use of luminescent nanoparticles emitting in the I-BW, exhibiting higher *S_rel_* and smaller *δT*, might improve the performance of the luminescent nanothermometers used up to now in cell thermometry. At the same time, the reduced light scattering in this spectral region would allow for a better spatial resolution, or at least better image definition, if thermal mapping in the I-BW is performed. In this context, the development of semiconducting polymers for imaging-guided photothermal therapy, using afterglow emissions pumped in the I-BW, is inspiring [52,53]. These semiconducting nanoparticles not only allowed the performance of ultrasensitive in vivo imaging by removing tissue autofluorescence, but they can also be used as afterglow luminescent thermometers and photothermal agents, being the first semiconducting polymer nanoparticles with this dual function. However, two different excitation sources had to be used, one for exciting the afterglow emissions that allowed the determination of temperature, and another one for promoting the release of heat, leaving room for future improvement of the material design.

## 4. Conclusions

In summary, we investigated the upconversion emissions generated by Tm,Yb:GdVO_4_@SiO_2_ core-shell nanoparticles, in the physiological range of temperatures, doped with different concentrations of Tm^3+^ after excitation at 980 nm. The two emission bands located at 700 and 800 nm, that lie in the I-BW, and arise from the thermally coupled ^3^F_2,3_ and ^3^H_4_ energy levels of Tm^3+^, respectively, allowed us to develop a luminescent thermometer operating via the *FIR* technique. The nanoparticles doped with 1 mol% Tm^3+^ exhibited the highest thermal absolute sensitivity for this purpose, with a maximum of 0.037 K^-1^ and an excellent thermal relative sensitivity of 3.3% K^−1^ at 333 K, which are within the highest values reported for Tm^3+^, Yb^3+^ co-doped systems, to the best of our knowledge. A penetration depth of 0.8 mm in biological tissues could be achieved by the 800 nm emission under excitation at 980 nm with a power of 50 mW. Furthermore, the inert silica shell surrounding the luminescent active core allowed the dispersal of the nanoparticles in biological compatible fluids like PBS, which facilitated their internalization in HeLa cells, proving their potential for biolabelling applications. Thus, Tm,Yb:GdVO_4_@SiO_2_ core-shell nanoparticles can be considered as a multifunctional platform for NIR-to-visible upconversion biolabeling, and NIR-to-NIR upconversion thermal sensing. This procedure allows a high thermal resolution, of the order of 0.15 K, resulting in the high thermal sensitivity achieved when compared with the performances of other luminescent thermometers operating in the visible range, traditionally used to perform cell thermometry. An additional advantage that should be explored in detail in the future is the lower spatial resolution that can be achieved in this spectral region due to the reduced light scattering. This would allow for discriminating, with a better accuracy, the differences in temperature and the thermal gradients generated within the organelles of living cells, and the cytoplasm, the nucleus and the membrane, permitting us to better understand the processes occurring inside the cells from a thermal point of view.

## Figures and Tables

**Figure 1 nanomaterials-10-00993-f001:**
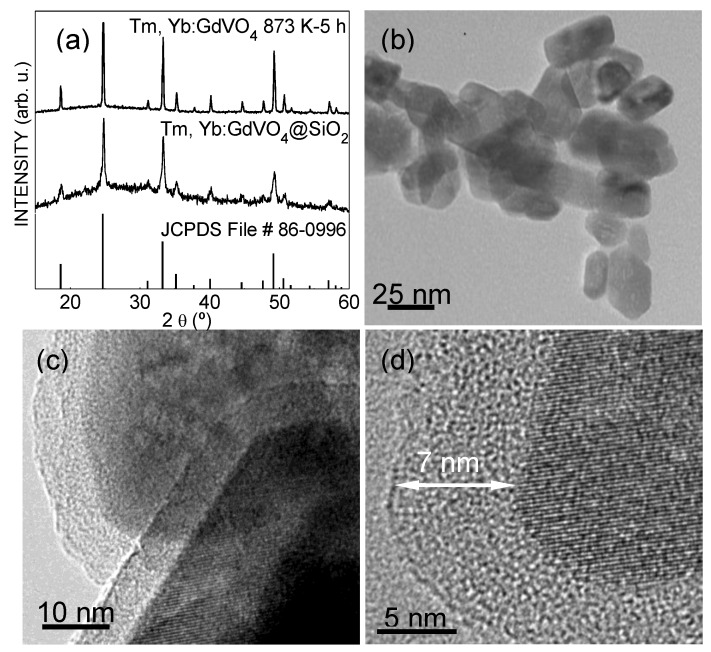
Structural and morphological characterization of Tm,Yb:GdVO_4_ nanoparticles. (**a**) X-ray diffraction patterns of the Gd_0.84_Yb_0.15_Tm_0.01_VO_4_ sample annealed at 873 K for 5 h, and coated with a 7 nm thick SiO_2_ layer. For comparison, the X-ray diffraction (XRD) pattern scheme of tetragonal *I*4_1_/*amd* GdVO_4_, Joint Committee on Powder Diffraction Standards (JCPDS) File 86-0996 has been also included. (**b**) Transmission electron microscope (TEM) image of the annealed Tm,Yb:GdVO_4_ nanoparticles (NPs). (**c**) and (**d**) Hig-resolution transmission electron microscope (HRTEM) images of core-shell Tm, Yb:GdVO_4_@SiO_2_ NPs, showing the lattice fringes of the crystalline core (darker areas) and the amorphous silica shell (lighter areas around the core).

**Figure 2 nanomaterials-10-00993-f002:**
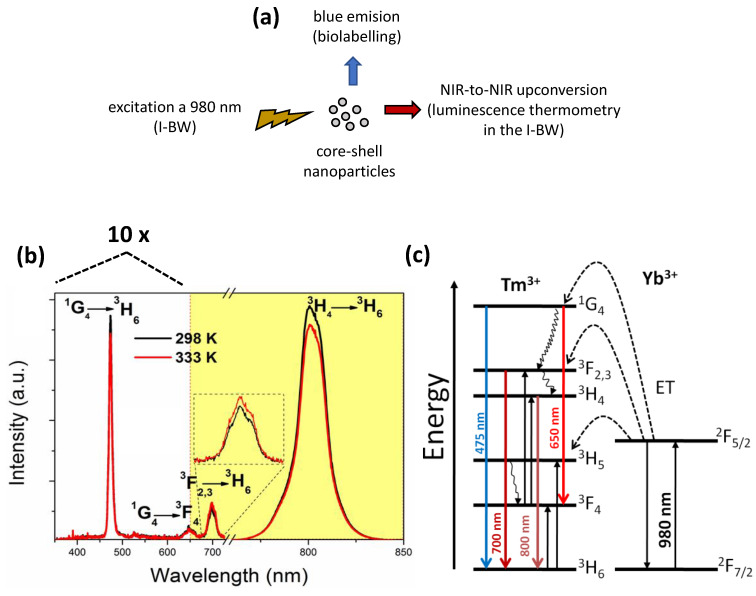
(**a**) Schematic representation of the pumping and emission bands generated by the Tm,Yb:GdVO_4_@SiO_2_ core-shell nanoparticles, to illustrate their practical use as biolabels, with emission in the blue, and luminescent thermometers operating in the first biological window (I-BW). (**b**) Upconversion emission spectra of the Tm,Yb:GdVO_4_@SiO_2_ core-shell nanoparticles at room temperature and at 333 K. Note that the visible part of the spectra, from 375 to 675 nm, has been magnified 10 times to respect the NIR part. The inset shows the magnification of the peak located at 700 nm, to make more evident the variation of the intensity of this peak with temperature. (**c**) Energy level diagram of Tm^3+^and Yb^3+^ ions in GdVO_4_, indicating the absorption, energy transfer and emission pathways.

**Figure 3 nanomaterials-10-00993-f003:**
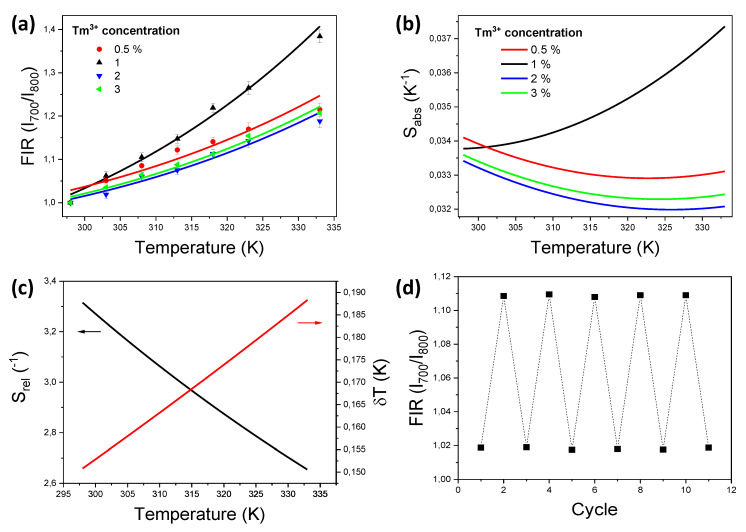
(**a**) Normalized fluorescence intensity ratio (*FIR)* as a function of temperature for the Tm,Yb:GdVO_4_@SiO_2_ core-shell nanoparticles doped with different concentration of Tm^3+^. (**b**) Absolute thermal sensitivity of the (0.5–3 mol%) Tm, Yb:GdVO_4_@SiO_2_ core-shell nanoparticles. (**c**) Relative thermal sensitivity and temperature uncertainty for Tm,Yb:GdVO_4_@SiO_2_ core-shell nanoparticles. (**d**) *FIR* recorded in 10 heating-cooling temperature cycles, showing a reproducibility higher than 99%.

**Figure 4 nanomaterials-10-00993-f004:**
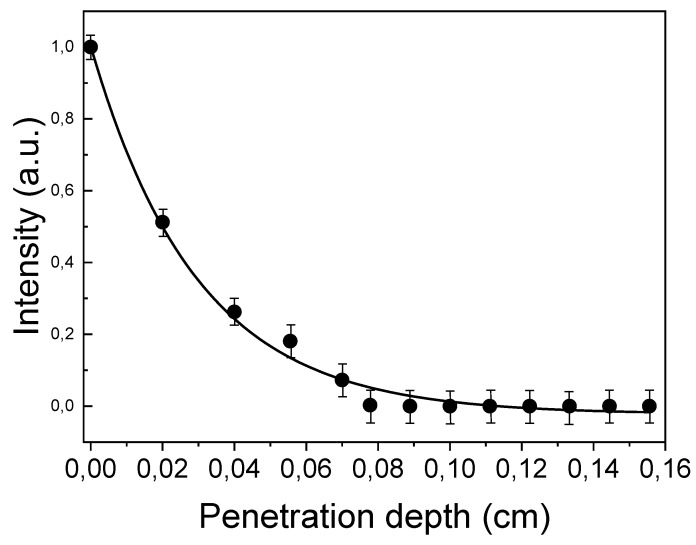
Normalized intensity of the upconversion emission at 800 nm, generated by Tm,Yb:GdVO_4_@SiO_2_ core-shell nanoparticles in a 2% intralipid aqueous solution to respect the intensity value measured at the surface of the phantom tissue as a function of the penetration depth (continuous line was done for guiding the eyes).

**Figure 5 nanomaterials-10-00993-f005:**
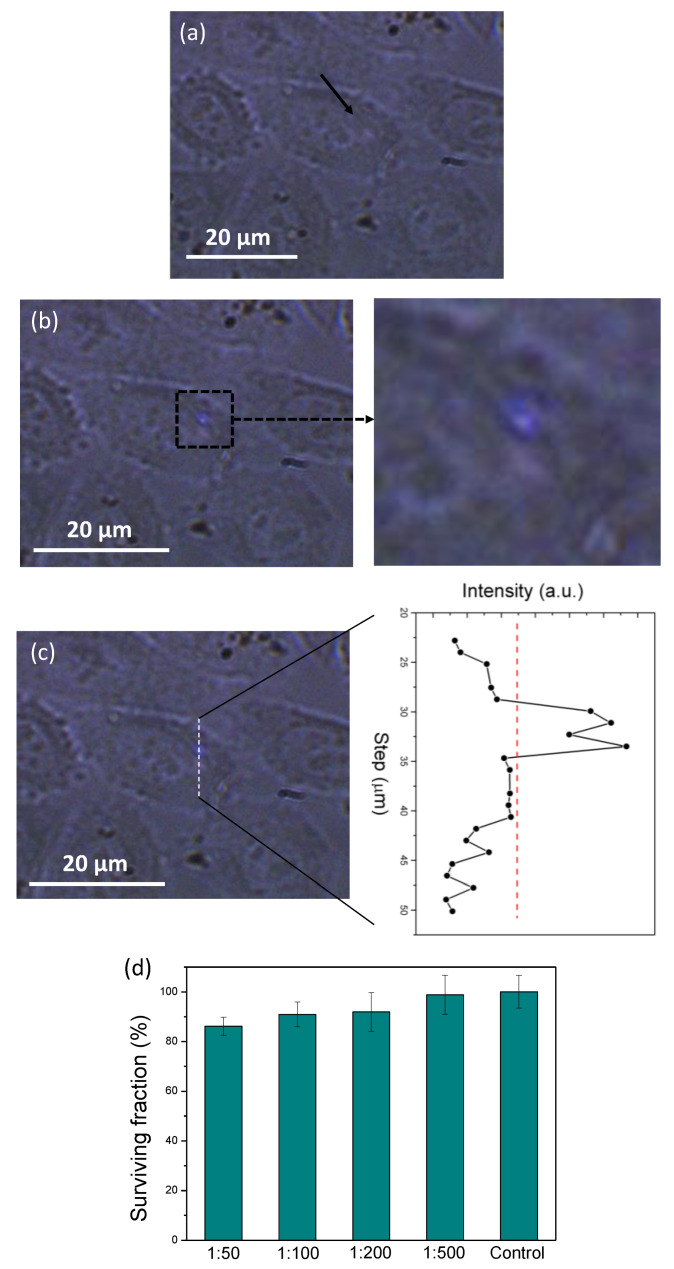
Microscope optical transmission images of the HeLa cells incubated with Tm,Yb:GdVO_4_@SiO_2_ core-shell nanoparticles when the excitation laser is (**a**) off, and (**b**) on. The arrow indicates the position were the core-shell nanoparticles are located, as can be seen when the laser is on in (**b**), together with an enlarged image of the square area indicated in the image. (**c**) Intensity scan of the blue emission along the dashed line indicated in the figure with a scan step of 2 μm. (**d**) Percent survival of HeLa cell line incubated with different concentrations of Tm^3+^,Yb^3+^:GdVO_4_@SiO_2_ core-shell nanoparticles. Each point corresponds to the mean value ± standard deviation from five different experiments.

**Table 1 nanomaterials-10-00993-t001:** Comparison of the performance of Tm^3+^-doped systems used in luminescence nanothermometry operating totally or partially in the I-BW.

Material	Temperature Range (K)	Excitation Wavelength (nm)	Wavelength Ratio	*S_abs_* (K^−1^) (T) (K)	*S_rel_* (% K^−1^) (*T*) (K)	*δ**T* (K)	Ref.
Tm,Yb:GdVO_4_@SiO_2_	298–333	980	I_700_/I_800_	0.037 (333)	3.3 (298)	0.15	This work
Tm, Yb:YVO_4_	288–473	980	I_800_/I_700_	-	2.4 (425)	~0.1	[34]
Tm,Yb:LiNbO_3_	323–773	980	I_700_/I_800_	0.00024 (773)	3.0 (323)	0.16 *	[19]
Tm,Yb:LiNbO_3_ (single crystal)	323–773	980	I_700_/I_800_	0.00033 (773)	3.0 (323)	0.16 *	[35]
Tm,Yb:LiNbO_3_ (polycrystal)	323–773	980	I_700_/I_800_	0.00024 (773)	2.0 (323)	0.25 *	[35]
Tm, Yb:YA Goptical fiber	333–733	976	I_683_/I_782_	0.021 (733)	2.3 (333)	0.22 *	[33]
Tm:NaYbF_4_@SiO_2_	100–700	980	I_697_/I_798_	0.0066 (100)	28 (100)	0.02 *	[36]
Tm:NaYbF_4_	298–778	980	I_700_/I_800_	0.0002 (778)	1.8 * (298)	1.5	[37]
Tm, Yb:KLuF_4_	303–503	980	I_690_/I_795_	-	1.0 (303)	0.5 *	[38]
Tm,Yb:YF_3_	300–750	980	I_700_/I_776_	0.001 (750)	2.8 * (300)	0.18 *	[39]
Tm,Yb:oxyfluoride glass	293–703	980	I_700_/I_800_	0.00029 (703)	3.2 * (300)	0.16 *	[40]
Tm,Yb:Y_2_O_3_	6–300	978	I_815_/I_454_	-	7.8 (270)	-	[41]
Tm,Yb:Y_2_O_3_	6–300	978	I_815_/I_460_	-	6.7 (178)	-	[41]
Tm,Yb:Y_2_O_3_	6–300	978	I_815_/I_656_	-	6.4 (290)	-	[41]

* Values calculated by us for this work, taking into account the values given by the authors in the original paper.

**Table 2 nanomaterials-10-00993-t002:** Cellular thermometry using lanthanide-doped materials.

Material	Thermometric Parameter	Cell Line	Introduction into Cells	Spectral Range (nm)	*λ_exc_* (nm)	*δ**T* (K)	Ref.
Eu-TTA	Emission intensity	CHO	Incubation, RT, 30 min On cell membrane	615	372	-	[46]
Eu-TTA	Emission intensity	HeLa	Incubation, 310 K, 1 h Endosome/lysosome			0.3	[47]
Eu-TTA/Rh101	Intensity ratio	HeLa	Incubation, 310 K, 2 h Endosome	580–615	340 (Eu-TTA) 530 (Rh101)	1.0	[48]
Eu-TTA and Fluo-4	Emission intensity	HeLa	Micropipette, physical contact	-	385 (Eu-TTA) 480 (Fluo-4)	<0.5	[49]
NaYF_4_:Er,Yb	*FIR*	HeLa	Incubation, 1.5 h	525–545	920	-	[50]
NaYF_4_:Er,Yb	Intensity ratio	HEK 293	Transfected	539–651	980 50 mW	-	[51]

TTA: thenoyltrifluoroacetonate; CHO: Chinese hamster ovary; HEK: human embryo kidney.

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
