# Peer review of "Bifunctional Tm3+,Yb3+:GdVO4@SiO2 Core-Shell Nanoparticles in HeLa Cells: Upconversion Luminescence Nanothermometry in the First Biological Window and Biolabelling in the Visible"

_nanomaterials, 2020, doi:10.3390/nano10050993_

Round 1
Reviewer 1 Report
The authors investigated the UC properties in the first biological window of Tm,Yb:GdVO4@SiO2 core-shell nanoparticles under 980 nm pumping, determined their thermal sensing properties and the penetration depth in biological tissues.
The Lanthanide-doped upconversion nanoparticles are widely studied for bioimaging applications:
- Nanotechnology Reviews Volume 6: Issue 2 (2016) Upconversion nanoparticles for bioimaging
- Recent Progress of Rare‐Earth Doped Upconversion Nanoparticles: Synthesis, Optimization, and Applications, TrAC Trends in Analytical Chemistry, Volume 120, November 2019, 115646
- Lab on upconversion nanoparticles: optical properties and applications engineering via designed nanostructure, Xiaomin Li et al Chem. Soc. Rev., 2015,44, 1346-1378
- Upconversion nanoparticles for in vivo applications: limitations and future perspectives Blanca del Rosal1,4 and Daniel Jaque, Methods and Applications in Fluorescence, Volume 7, Number 2
And many, many others!
In particular, the UC properties of the material for thermal sensing and the role of SiO2 coating was deeply studied by the authors:
- Savchuk, O.A.; Carvajal, J.J.; Haro-Gonzalez, P.; Aguilo, M.; Diaz, F. Luminescent nanothermometry using short-wavelength infrared light. J. Alloys Compnd. 2018, 746, 710-719, 530
- Savchuk, O.A.; Carvajal, J.J.; Cascales, C.; Aguiló, M.; Díaz, F. Benefits of Silica Core−Shell Structures on 515 the Temperature Sensing Properties of Er,Yb:GdVO4 Up-Conversion Nanoparticles. ACS Appl. Mater. 516 Interfaces 2016, 8, 7266-7273, https://doi.org/10.1021/acsami.6b01371.
- Nisrin Mohamed Bhiri, M.Dammak, Magdalena Aguiló, Francesc Díaz Joan J.Carvajal, M.C.Pujol Stokes and anti-Stokes operating conditions dependent luminescence thermometric performance of Er3+-doped and Er3+, Yb3+ co-doped GdVO4 microparticles in the non-saturation regime Journal of Alloys and Compounds, volume 814, 25 January 2020, 152197
- Calderon-Villajos, R.; Zaldo, C.; Cascales, C. Enhanced upconversion multicolor and white light luminescence in SiO2-coated lanthanide-doped GdVO4 hydrothermal nanocrystals. Nanotechnol. 2012, 23, 545 505205, https://doi.org/10.1088/0957-4484/23/50/505205
Therefore, the authors have to clarify/indicate the novelty and the contribution to the development of areas of current scientific interest
2. The authors stated that “These results probe that Tm,Yb: GdVO4@SiO2 core-shell nanoparticles can be used as thermal probes operating in the I-BW with a very good thermal sensitivity and thermal resolution, and with possibilities to determine the temperature inside living HeLa cells in the future.” but without any attempt to do that. …despite the good performances of the materials. Why? In my opinion this point would have been the most important one of their study!
Reviewer 2 Report
In this work, Savchuk and co-workers have developed novel lanthanide-based core-shell Upconversion Nanoparticles (UCNPs), toward their use as a nano-thermometer sensitive to the physiological range of temperatures. The authors also tried to characterize the properties of these UCNPs in vitro in Hela cells. This is an interesting study with a relevant aim, being a good fit for Nanomaterials.
Nevertheless, there are some aspects that must be improved by the authors, before recommendation for acceptance can be given. Namely:
-The quality of used English must be significantly improved;
-The authors state that they have used a environmentally friendly soft hydrothermal route for the preparation of the proposed nanoparticles (lines 105-106). However, this statement is not validated anywhere in the manuscript. The authors must then validate the greenness (or not) of this procedure by using available metrics, such as EcoScale, E-factor, cEF and/or PMI (see DOIs: 10.1039/c9gc01958h; 10.3762/bjoc.13.206), or even perform life cycle assessment studies (see DOIs: 10.3390/ma13030504; 10.3390/ma13071487);
-The authors must provide the synthesis yield (w/w, in %) of the Gd0.85-xYb0.15TmxVO4 nanoparticles;
-How many times was the synthesis of the proposed nanoparticles repeated to ensure that different batches generate identical nanomaterials with reproducible properties?
-If I am not mistaken, while the authors claim to develop a novel nano-thermometer that could be used in the physiological range of temperatures, there was no effort to demonstrate that the proposed nanoparticles have such thermal sensitivity inside cells;
-This paper lacks more information regarding the internalization degree of the nanoparticles, and regarding their cellular localization;
-The authors checked the "dark" cytotoxicity of the nanoparticles, which showed some degree of biocompatibility. However, the authors must also check the cytotoxicity of the nanoparticles after NIR irradiation, to guarantee that these nanoparticles do not induce any photothermal/photodynamic effect that causes damage to the cells, and so, negates their usefulness as a bioimaging probe;
Round 2
Reviewer 1 Report
The authors gave proper answer/comments to to questions raised.
Reviewer 2 Report
The authors have addressed all my comments, and so, recommendation for acceptance can be given.